# A prospective cohort analysis from Germany shows transition into adulthood is an underestimated vulnerable period for children with overweight/obesity

## Abstract

**Background** Even though the transition from adolescence to adulthood is a vulnerable and crucial period for young individuals with chronic conditions, it remains poorly studied in obesity. We comprehensively characterized the transition of children with overweight/obesity from childhood to adult care.
**Methods** Data from an adulthood follow-up of the Leipzig childhood obesity cohort (N = 209, mean age at follow-up 24.9 years) were analyzed and related to the respective childhood data. Assessments comprised anthropometrics, oral glucose tolerance testing, carotid ultrasonography, 24 h blood pressure monitoring, liver elastography and questionnaires on psychosocial background, quality of life and transition process.
**Results** Here we show that childhood onset overweight/obesity persists in 93.3% until adulthood. However, most patients (55.6%) who move into a higher BMI category during transition still report that things are going better with their weight during adulthood. Complications are frequent among young adults (overall 83.7%, (pre)diabetes 11.4%, hypertension 24.5%, elevated intima-media thickness 82.2%, fatty liver disease 10.2%, regular intake of medication 45%) and deteriorate with progressing weight gain from childhood to adulthood. 87% of adults have not been aware of one or more of these complications before. Most participants report physical complaints (57.3%) and symptoms of depression/anxiety (52.0%). The majority (94.3%) lose contact with specialized obesity care during transition with lack of knowledge being the most frequent reason. Instead, general practitioners are regularly consulted by 50.7%, but obesity is seldom addressed as a topic (22.6%). Better management of the transition process is desired by 26.7% of participants.
**Conclusions** Children with overweight/obesity carry a substantial health burden into adulthood which is often not perceived. Structured transition approaches are needed to address obesity as a chronic condition.

## Plain language summary

Young people with obesity often continue to have weight issues into adulthood, however they have been poorly followed so far. In our study, we conducted a follow-up of children with obesity at young adulthood. We revealed that health complications such as diabetes, high blood pressure, and fatty liver disease are common, yet many remain unaware. Many also experience physical discomfort and mental health struggles. Most lose contact with obesity specialists, relying instead on general doctors, who rarely discuss obesity. A quarter of participants wanted better support during the transition to adulthood. The study highlights the need for structured programs to help manage obesity as a long-term condition.

Obesity is a chronic health condition, and its prevalence has increased worldwide both among adults and children[1,2]. In Germany, 15.4% of children and 53.5% of adults experience overweight or obesity[3,4]. If obesity manifests during childhood it is very likely to persist into adulthood[5–7], hence it should be regarded as a chronic disease. Childhood onset obesity causes premature mortality in adulthood by increasing rate of complications such as type 2 diabetes mellitus (T2D) and cardiovascular diseases[8–13].

Moreover, those complications deteriorate faster if they commence during childhood[14].

A critical period for adolescents with special healthcare needs is the shift from pediatric care to adult medical care, which is referred to as the transition[15]. It poses a variety of challenges, including separation from the parental home, establishing new social relationships, developing and implementing own world views and establishing responsibility within the healthcare

system[16–18]. Emerging evidence suggests that health of young people with chronic conditions worsens during the transition period. For instance, 40% of adolescent kidney transplant recipients lost their transplants within 36 months after transitioning, primarily due to inadequate therapy adherence[19,20]. Likewise, 40% of adolescents with type 1 diabetes lost contact with specialized care, leading to a significantly increased risk for poor glycemic control[21], and young adults with T2D experienced deteriorating glycemic control and loss to follow-up during the transition period[22].

However, transition in the context of obesity as a chronic condition has been neglected so far. Policies, clinical guidance and expert opinions are lacking internationally, putting the transition of childhood obesity into the focus of current clinical research[23].

In this study, we adopt a comprehensive approach to describe the transition of children and adolescents with overweight or obesity into adulthood from an obesity center in Germany. We demonstrate that childhood onset overweight/obesity persists in most cases (93.3%) with a high rate of physical complaints (57.3%) and complications (83.7%) already in early adulthood, which is only partially perceived by the patients. Given the high disease burden in this population, it is especially alarming that many of those adults with childhood onset obesity loose contact to the healthcare system as transition is accompanied by structural changes. Collectively, these results support the need for structured transition approaches to address obesity as a chronic condition.

## Methods
### Study population and design
Data were retrieved from the Leipzig childhood obesity cohort (NCT04491344[24]). Children and adolescents from the area around Leipzig were referred to the obesity outpatient clinic of Leipzig University Medical Center for evaluation and treatment of overweight/obesity and included in the study with subsequent regular follow-ups.

Participants who reached adulthood were followed up between July 2021 and March 2023, facilitated by the clinical trial unit of the Helmholtz Institute for Metabolic, Obesity and Vascular Research (HI-MAG) of the Helmholtz Zentrum München at the University of Leipzig. The study was approved by the institutional review board of the Medical Faculty of Leipzig University (Reg. No. 007/04-ek). Informed written consent was obtained prior to study participation from all participants aged 12 years and older and from their legal guardian if they were younger than 18 years of age. Only participants without syndromic or secondary obesity were included ($N = 1231$) resulting in 209 subjects for further analysis (Fig. 1 and Table 1). Of note, patients who were not willing to participate after being successfully recontacted had a similar age- and BMI distribution at their last childhood visit than those who finally consented to the follow-up examination and slightly more men than women dropped out of the study (Table S1). Adult participants unable or unwilling to attend an on-site visit at the HI-MAG clinical trial unit were asked to take an online survey remotely. In total, $N = 135$ subjects attended an on-site visit and $N = 74$ individuals completed the online survey. For comparison of the adult follow-up cohort with the respective childhood data, we utilized the latest visit during childhood (Table 1).

### Anthropometric assessment
Height and weight were collected by trained staff members and were rounded to the nearest decimal place. For online participants at follow-up, values are based on self-assessment. For childhood data, the body mass index (BMI) was transformed to age- and sex-specific standard deviation scores (SDS) and percentiles according to the German population[25]. Normal weight was defined as a BMI between the 10th and 90th percentile, overweight from the 90th to the 97th percentile, obesity from the 97th percentile to the 99.5th percentile and extreme obesity encompassing values > 99.5th percentile. In adulthood, overweight was defined as a BMI ranging from 25–29.9 kg/m², obesity class 1 from 30–34.9 kg/m², obesity class 2 from 35–39.9 kg/m² and obesity class 3 as a BMI ≥ 40 kg/m² according to the World Health Organization's (WHO) Obesity Classification[26]. For BMI

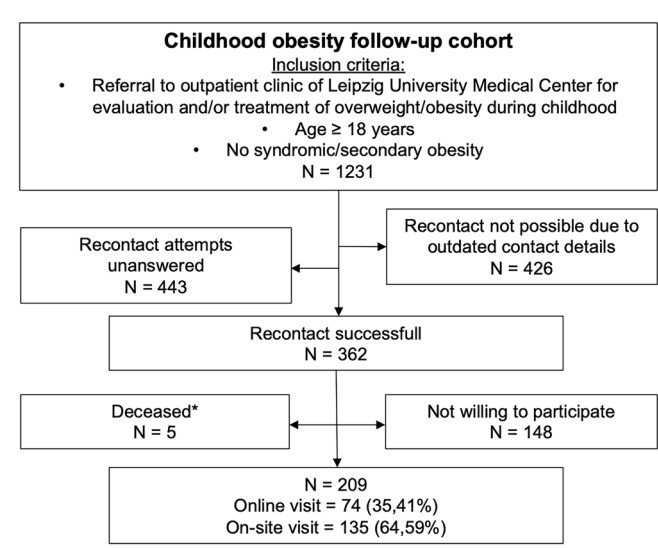

**Fig. 1 | Selection of study population.** *Reasons for death: Heart attack ($N = 1$), Epileptic seizure ($N = 1$), Car accident ($N = 1$), Unknown ($N = 2$). *N* number of subjects.

comparison, 'Obesity' in childhood corresponded to 'Obesity class 1' in adulthood and 'Extreme obesity' in childhood corresponded to 'Obesity class 2 or 3' in adulthood.

### Laboratory assessments
Laboratory assessments were analyzed by the local laboratory of the Leipzig University Medical Center from a venous blood sample drawn after an overnight fast, as described previously, according to the laboratory standards[24]. For Oral Glucose Tolerance Testing (OGTT), glucose and insulin levels were measured for 2 h after oral intake of 1.75 g/kg body weight of dextrose (maximum 75 g). Childhood glucose levels were measured in hemolysates with an enzymatic-amperometric method (Super GL Speedy, Dr. Müller Gerätebau GmbH, Freital, Germany) and during adulthood in plasma by a Cobas 8000 analyzer (Roche Diagnostics, Mannheim, Germany).

Prediabetes or diabetes was diagnosed according to the ADA guideline[27] if at least two of three diabetes markers (fasting glucose (FG), 2-h glucose, HbA1c) were elevated. The prediabetes range comprises FG levels of 5.6–6.9 mmol/l, 2-h glucose from 7.8–11.0 mmol/l, and HbA1c levels of 5.7–6.4%. The diabetic range includes FG levels of ≥7.0 mmol/l, 2-h glucose values of ≥11.1 mmol/l, and HbA1c levels of ≥6.5%. Patients with the intake of antidiabetics or an established diabetes diagnosis were also considered pathological. Normal glucose metabolism was considered if none of the three diabetes markers were elevated.

### Questionnaires
Psychosocial aspects were assessed using the standardized questionnaire EQ-5D-5L[28]. Questions on transition success and self-responsibility in adult care were partly adapted and customized from a 'Transition readiness assessment' questionnaire from 'The National Alliance to Advance Adolescent Health' sponsored 'Got Transition' program[29]. Additional questions (Supplementary Data 1) were developed by the research team through consensus of three pediatric experts.

### Evaluation of fatty liver disease
Vibration-controlled transient liver elastography was facilitated by the FibroScan® 502 TOUCH device (Echosens SA, Paris, France) according to the manufacturer's instructions. Based on ultrasound waves, liver stiffness measurement (LSM) was derived by the propagation of produced shear waves. The measurement was successful if 10 valid data points could be measured with a variance below 30%. Fatty liver disease was suspected if

**Table 1 | Cohort characterization**

| | | Childhood | Adulthood |
|---|---|---|---|
| Form of participation | N (%) on-site | 209 (100%) | 135 (64.59%) |
| | N (%) online | – | 74 (35.41%) |
| Age in years | Mean (range) | 13.24 (2–17) | 24.91 (18–36) |
| Sex | N (%) male | 71 (33.97%) | 71 (33.97%) |
| | N (%) female | 138 (66.03%) | 138 (66.03%) |
| BMI in kg/m$^2$ | Mean (range) | 31.41 (20.52–65.14) | 36.18 (19.73–80.0) |
| BMI SDS | Mean (range) | 2.59 (0.91–5.03) | – |
| BMI category | N (%) Normal weight | 2 (0.96%) | 14 (6.70%) |
| | N (%) Overweight | 25 (11.96%) | 48 (22.97%) |
| | N (%) Obesity/Obesity class 1[a] | 85 (40.67%) | 49 (23.44%) |
| | N (%) Extreme Obesity/Obesity class 2[a] | 97 (46.41%) | 32 (15.31%) |
| | N (%) Extreme Obesity/Obesity class 3[a] | | 66 (31.58%) |
| Glucose metabolism[b] | N (%) Normal | 115 (85.19%) | 100 (74.07%) |
| | N (%) Prediabetes | 18 (13.33%) | 5 (3.7%) |
| | N (%) Diabetes | 2 (1.48%) | 10 (7.42%) |
| Fatty liver disease[b] | N (%) Normal | 72 (86.75%) | 115 (89.84%) |
| | N (%) Elevated liver enzymes | 11 (13.25%) | 11 (8.59%) |
| | N (%) Elevated liver enzymes + pathological liver elastography | – | 2 (1.57%) |
| Blood pressure[b] | N (%) Normal | 59 (45.04%) | 88 (65.19%) |
| | N (%) Elevated | 11 (8.04%) | 14 (10.37%) |
| | N (%) Hypertension | 61 (46.92%) | 33 (24.44%) |
| Intima-media thickness (IMT)[b] | N (%) Normal | – | 16 (12.40%) |
| | N (%) IMT ≥ P75 | – | 106 (82.17%) |
| | N (%) Plaque and IMT ≥ P75 | – | 7 (5.43%) |
| How many drugs do you take regularly? | N (%) 0 | – | 114 (55.07%) |
| | N (%) 1 | – | 39 (18.84%) |
| | N (%) 2 | – | 22 (10.63%) |
| | N (%) 3 | – | 14 (6.76%) |
| | N (%) 4 | – | 11 (5.31%) |
| | N (%) 5+ | – | 7 (3.38%) |

Of note, two participants had a normal BMI SDS at their last childhood visit, even though they had experienced overweight or obesity in previous recordings and therefore had consulted the obesity outpatient clinic.

N number of subjects, P75 75th percentile.

[a]"BMI class 1–3" is only applicable for adulthood, whereas the categories "Obesity" and "Extreme obesity" are only applicable for childhood.

[b]Data refers only to on-site participants.

ALT or AST were elevated and/or elastography revealed LSM values ≥ 8 kPa according to current German guidelines[30].

### Blood pressure assessment

Evaluation of arterial hypertension during adulthood was based on 24-h ambulatory blood pressure monitoring (ABPM) following the 2018 ESC/ESH guideline[31]. Only ABPM recordings with ≥70% valid recordings, ≥20 valid daytime, and ≥7 nighttime recordings were included. Mean systolic values > 130 mmHg were considered as pathological. If ABPM data were unavailable, the diagnosis relied on the mean value of three repetitive office measurements (normal <130 mmHg, elevated 130–139 mmHg, and hypertensive ≥140 mmHg)[31]. Intake of antihypertensive medication and/or prior diagnosis of arterial hypertension was also considered.

Blood pressure during childhood was assessed based on office measurements according to the national German guideline[32], considering age-, sex- and height-specific percentiles. Systolic blood pressure was classified as normal (<90th percentile), elevated (90th–94th percentile) or hypertensive (≥95th percentile).

### Intima-media thickness (IMT) measurement

IMT measurement in adult participants was conducted according to the Mannheim Intima-Media Thickness Consensus[33]. Patients were positioned supine with simultaneous ECG recording. IMT was evaluated on the distant wall of the common carotid artery at least 5 mm below the bulb along a minimum of 10 mm length of an arterial segment with an automated edge detection system at the end of diastolic movement. The mean of 3 repetitive measurements was calculated. According to current guidelines[34], IMT was considered elevated if it exceeded the age- and sex-specific 75th percentile. Plaque was defined as a focal structure encroaching into the arterial lumen of at least 0.5 mm or 50% of the surrounding IMT value.

### Statistical analysis

Descriptive statistical analysis was conducted using Microsoft Excel. All figures and graphs were generated using GraphPad Prism 10.2.2. Statistical differences between categorical variables were tested through $X^2$ test and differences between numerical variables through Student's t test after checking for normal distribution through visual inspection.

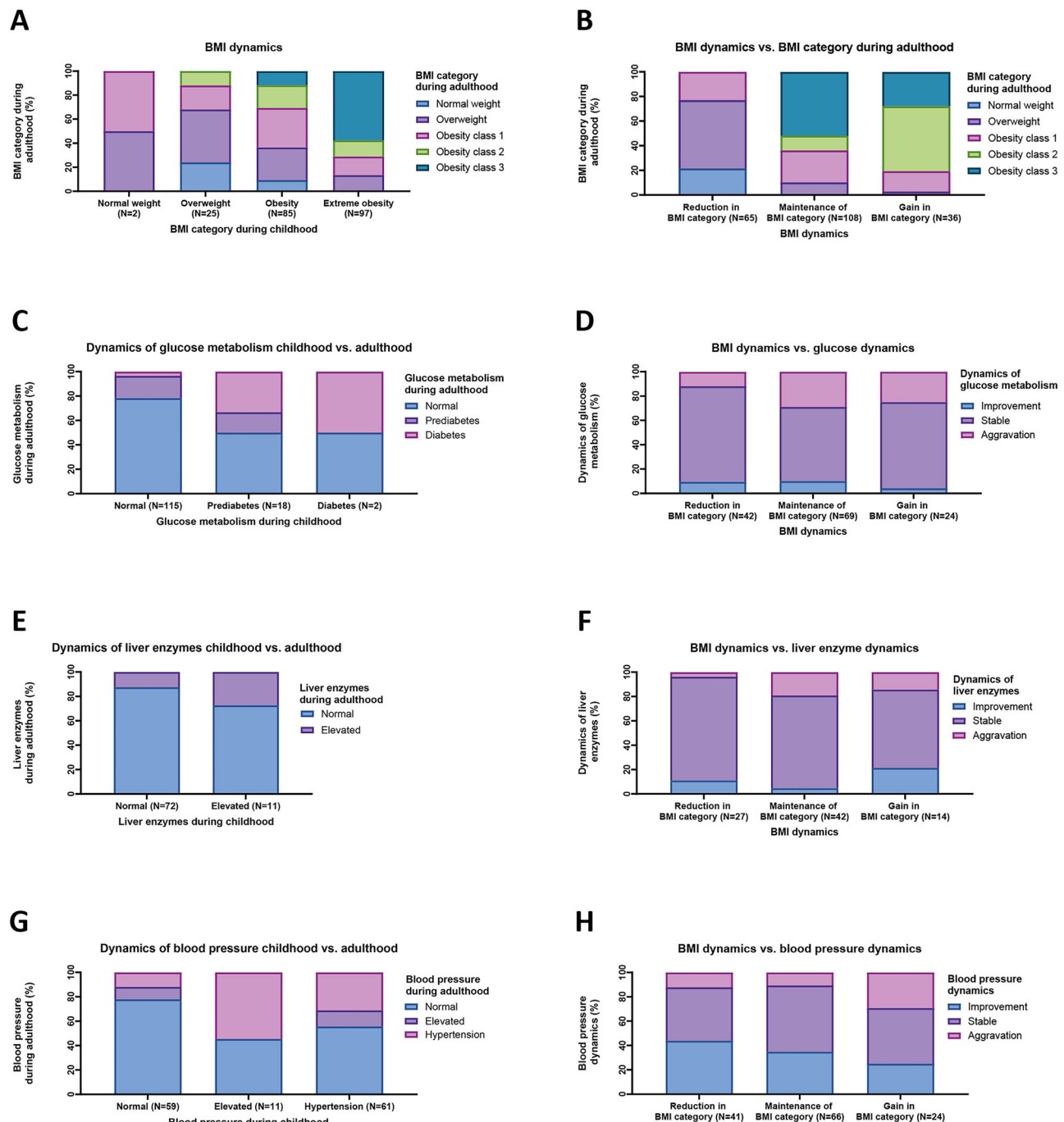

**Fig. 2 | Dynamics of health burden from childhood to adulthood.** We compared the health status from the last childhood visit with the follow-up at adulthood regarding BMI dynamics (**A, B**, N = 209), glucose metabolism (**C, D**, N = 135), fatty liver disease (**E, F**, N = 83) and blood pressure (**G, H**, N = 131). Of note, two participants had a normal BMI SDS at their last childhood visit, even though they had experienced overweight or obesity in previous recordings and therefore had consulted the obesity outpatient clinic (**A, B**). N number of subjects.

### Role of the funding source

The funders of the study had no role in study design, data collection, data analysis, data interpretation, or writing of the report.

### Results

We assessed the health burden of young adults with childhood-onset obesity and compared our results with the perceived health burden. Based on those findings, we looked at structural changes and challenges that patients were facing and how those could have been overcome.

### Health burden of young adults with childhood-onset obesity

We conducted a follow-up of 209 patients (two thirds female) who attended the obesity outpatient clinic of Leipzig University Medical Center during childhood and consented to a follow-up examination at adulthood, accompanying their transition up to a maximum age of 36 years (Table 1 and Fig. 1). Childhood onset overweight/obesity persisted in nearly all participants (93.3%), with class 3 obesity being the most prevalent (31.6%), nearly one-fourth had overweight and only a small percentage (6.70%) fell within the normal weight range (Table 1). Half of the participants (51.7%) maintained their BMI category

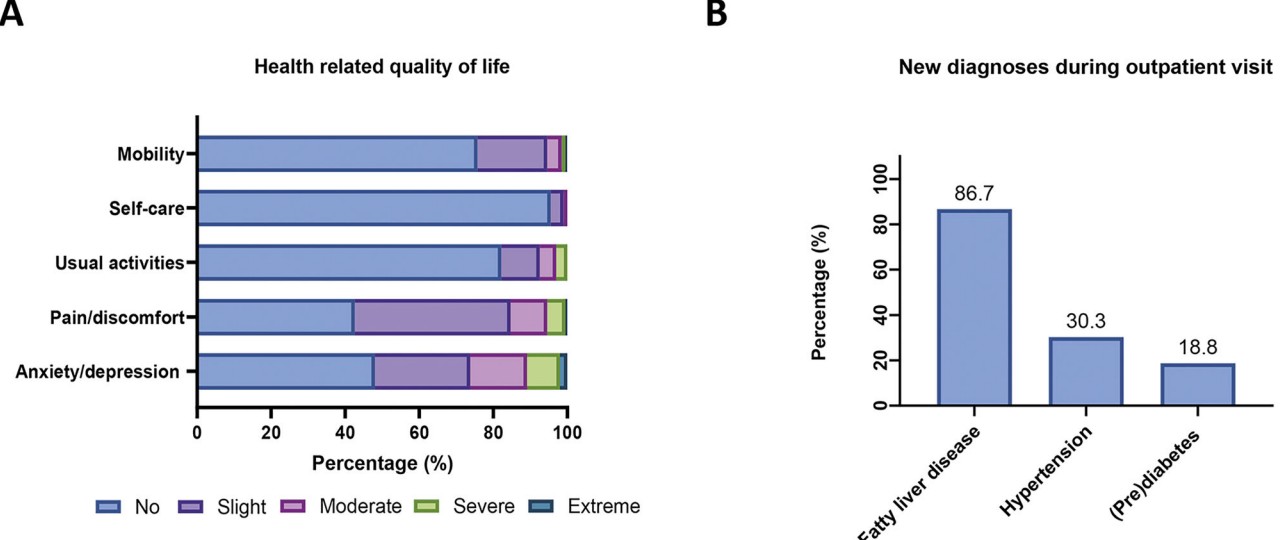

**Fig. 3 | Perceived health burden of young adults with childhood onset obesity.** **A** illustrates the responses to the EQ-5D-5L questionnaire (N = 202); **B** refers to on-site participants only and depicts the shares of newly diagnosed complications in relation to the total number of participants bearing the respective disease (fatty liver disease N = 15, hypertension N = 33, (pre)diabetes N = 16). N number of subjects.

during transition from childhood to adulthood, whereas one third (31.1%) reduced their BMI category and 17.2% had a gain in BMI category. Nearly three quarters of children with extreme obesity still had obesity class 2 or 3 during adulthood and most children who reduced their BMI category reached overweight but not normal weight during adulthood (Fig. 2A, B).

Regarding glucose metabolism after transition, more than one quarter of on-site study participants exhibited abnormal diabetes markers (Table 1), with 7.4% having overt diabetes. Among individuals with normal glucose metabolism in childhood, 18.3% transitioned to prediabetes and 3.5% developed diabetes in adulthood (Fig. 2C). Likewise, half of the children with prediabetes remained prediabetic or progressed to diabetes in young adulthood. Of note, deterioration in glucose metabolism occurred more frequently among participants with maintenance or gain in BMI category than in those with a reduction in BMI category (29% and 25% respectively vs. 11.9%, Fig. 2D).

Elevated ALT and/or AST enzyme levels as a proxy for hepatic steatosis were detected in 10.2% of young adults (Table 1). Furthermore, in 1.56% of subjects simultaneous elevation of liver enzymes and liver elastography suggested a high risk for advanced liver disease such as liver fibrosis (Table 1). Likewise, no liver fibrosis was detected by the non-invasive surrogate score Fib4 > 1.3[30]. Whereas one out of four children with elevated liver enzymes showed persistently high levels during adulthood, 72.2% reverted to normal levels (Fig. 2E).

Arterial hypertension was exhibited in nearly one-quarter of young adults and elevated blood pressure in 10.4% (Table 1). Among those with normal childhood blood pressure, more than one-fifth developed elevated blood pressure or hypertension in adulthood (Fig. 2G) and 31.2% of children with hypertension remained hypertensive in adulthood, whereas 55.7% reverted back to normal levels. We observed the strongest improvement of blood pressure in the group with a reduction in BMI category (43.9%) (Fig. 2H). Likewise, the highest aggravation occurred in those with a gain in BMI category (29.2%). Moreover, the vast majority of participants (82.2%) exhibited subclinical atherosclerosis during adulthood as represented by pathological IMT measurements and carotid plaque formation was detected in 5.4% of participants (Table 1).

The majority of participants (83.7%) reported at least one concomitant health condition (Fig. S1A, B) with regular intake of medication in 45% of cases (Table 1), again indicating a high chronic health burden within this population. Notably, 5 participants from the childhood obesity cohort were lost to follow-up due to premature death (1 heart attack, 1 epileptic seizure, 1 car accident, 2 unknown; Fig. 1).

## Perceived and undetected health burden of young adults with childhood-onset obesity

Next, we tested whether the high and chronic health burden of young adults with childhood-onset obesity is perceived adequately by the patients. When asked about their health-related quality of life, most participants (57.3%) reported daily pain/discomfort and 52% reported symptoms of anxiety/depression (Fig. 3A). In contrast, only 4.3% of participants stated depression as a concomitant disease (Fig. S1). Moreover, one out of four subjects experienced limitations in mobility, and 17.8% reported limitations in usual activities (Fig. 3A). Participants rated their overall health as moderately diminished, with a mean score of 74.4 (±16.2) out of 100 on a visual scale ranging from 0 (worst) to 100 (best).

Still, most participants (69.5%) felt more content with their weight during adulthood than during childhood (Fig. S2A). Surprisingly, even among participants with a gain in their BMI category, the majority (55.6%) reported that "things went better with their weight" during adulthood (Fig. S2B). In line, a substantial portion of young adults did not know about a serious health condition that was newly diagnosed at the outpatient clinic visit: 30.3% of subjects with arterial hypertension, 86.7% of subjects with fatty liver disease and 18.8% of participants with (pre)diabetes were not aware of this before (Fig. 3B).

## Treatment continuation after transition

Most children with overweight or obesity (58.9%) stayed in pediatric care until coming of age (Fig. S3). Subsequently, half of participants (50.7%) were regularly seen by a general practitioner (GP) (Fig. 4A) and 93.4% of young adults had a GP but without regular appointments. Almost one-third did not see a doctor routinely (47.9% of men vs. 21.7% of women, Fig. S4) and only a small minority (5.7%) continued visits at the obesity outpatient clinic (Fig. 4A). Still, participants with regular intake of medications (N = 92) reported on average a good adherence with 80.4% stating they miss a dose maximum once per week (Fig. S5). Of note, patients with maintenance or a gain in their BMI category from childhood to adulthood were more likely to see a doctor on a regular basis than those with a reduction in BMI category (Table S2).

Considering the potential key role of GPs in treating obesity in young adults, participants were asked how often weight was addressed during doctor appointments and only 22.6% experienced regular or frequent conversations about their weight (Fig. 4B). Among participants who discontinued specialized obesity outpatient care during adulthood, the most

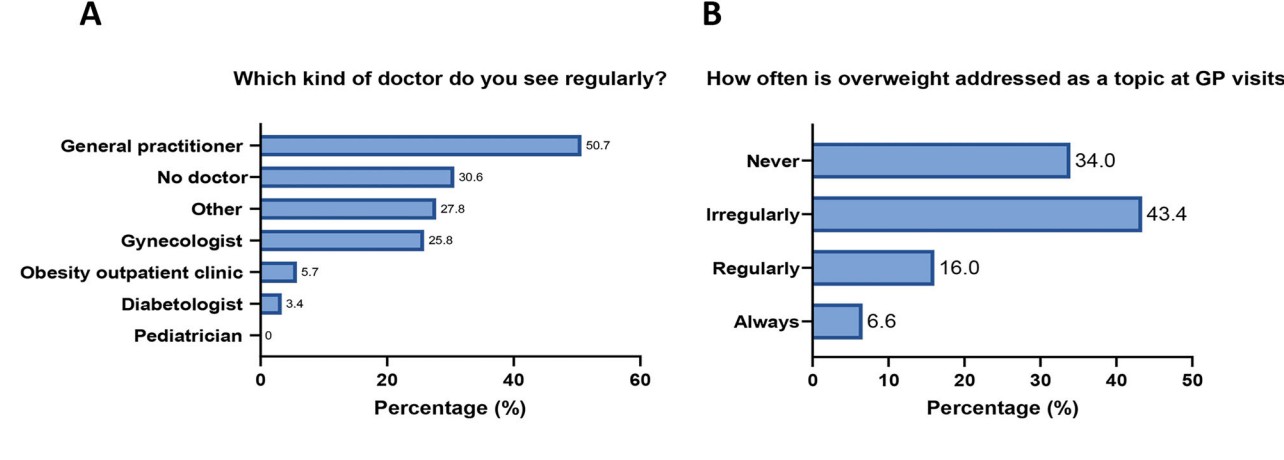

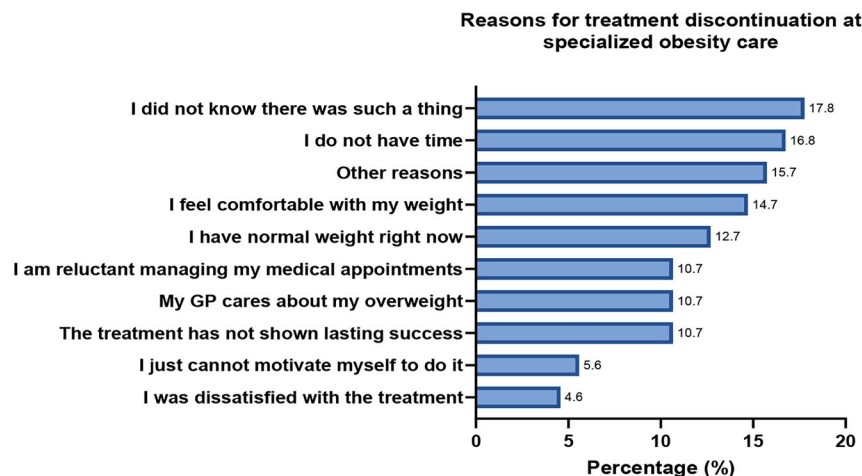

**Fig. 4 | Treatment continuation after transition from pediatric obesity care.** **A** refers to all participants ($N = 209$). Only participants who were still seeing a general practitioner (GP) on a regular basis were included in (**B**) ($N = 106$). Only participants who discontinued specialized obesity treatment were included in (**C**) ($N = 197$). Multiple answers were possible for (**A**) and (**C**). N number of subjects.

common reasons comprised unawareness of this possibility (17.8%), lack of time (16.8%) and feeling comfortable with the present weight (14.7%; Fig. 4C).

Most young adults reported at least one weight loss attempt in the past (92.3%, Table S3) with dietary change on participants' own initiative being the most prevalent one (66.0%). Out of those, 37% achieved a reduction in BMI category (Table S3). Every fourth in our cohort (26.3%) had participated in a structured lifestyle intervention program in the past. Out of those, most participants maintained their BMI category (52.7%) and 32.7% reduced their BMI category (Table S3). Participants with bariatric surgery ($N = 8$) either reduced ($N = 2$) or maintained ($N = 6$) their BMI category. All of them were seen by a doctor on a regular basis still during adulthood, 5 participants continued to receive specialized obesity care and 5 were seen by their GP. Participants on treatment with a GLP-1 receptor agonist had mixed weight-loss responses with most of them maintaining their BMI category ($N = 3$ out of 5). Again, all of them were seen by a doctor on a regular basis, either by a GP ($N = 3$) and/or a diabetologist ($N = 3$).

### Actual and desired transition from pediatric to adult medical care

After transition to adult medical care, most young adults reported commendable levels of self-responsibility and independence in managing their healthcare (Fig. 5). The majority agreed completely that they have knowledge regarding their medical conditions (63.4%), medications (83.7%) and allergies (69.3%). Moreover, 48% actively engaged in their healthcare by preparing questions in advance of doctor's appointments and 83.7% knew how to reach their healthcare provider when needed. However, 42.9% faced initial challenges in navigating the adult healthcare system, and 18.2% still relied on their parents for scheduling doctor's appointments (Fig. 5A, B).

Although 65.7% experienced a lack of specific transition support (Table 2), the majority of participants (73.3%) did not express the need for improvement when being asked "Would you have preferred a better transition?". Among those who did, the most common suggestions were providing contact information for adult medical care and joint consultations with the pediatrician and adult physician (Table 2).

## Discussion

We were able to demonstrate that children with overweight/obesity carry a substantial health burden into adulthood, which is only partially perceived by the patients. Given the high disease load in this population, it is especially alarming that many of those adults with childhood onset obesity lost contact to the healthcare system as transition is accompanied by structural changes.

In our study, the persistence of overweight or obesity from childhood to adulthood was high (93.3%), although 92% reported past weight loss attempts, supporting previous findings on the chronic nature of obesity[5,6]. Of note, the KiGGS study, as a nation-wide German follow-up[35], reported slightly less obesity persistence rates between 37% to 50% among adults aged

**A**

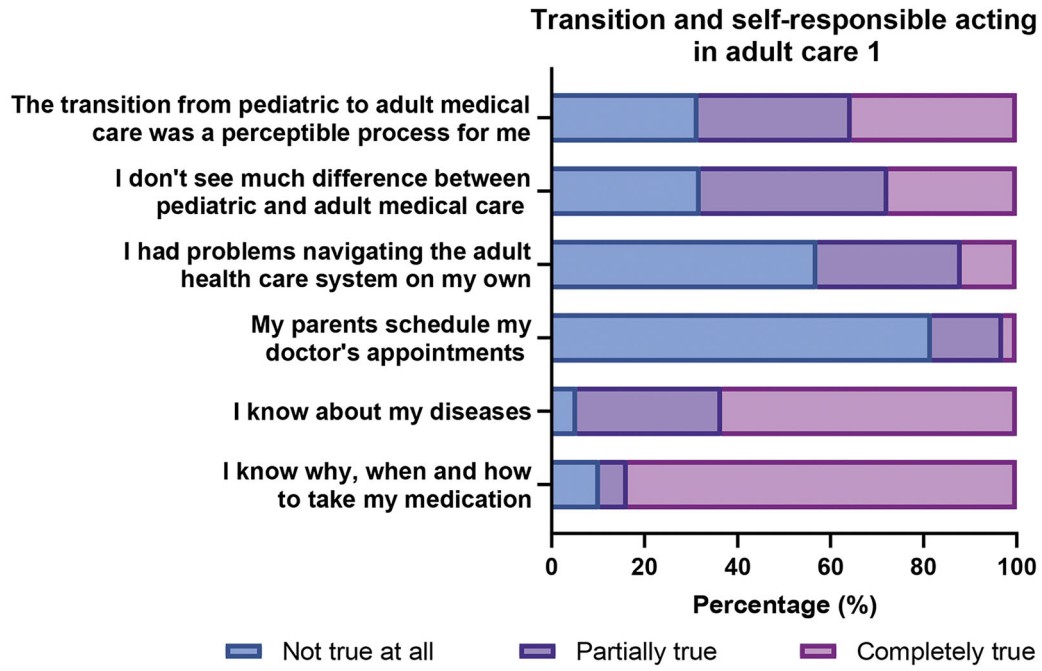

**B**

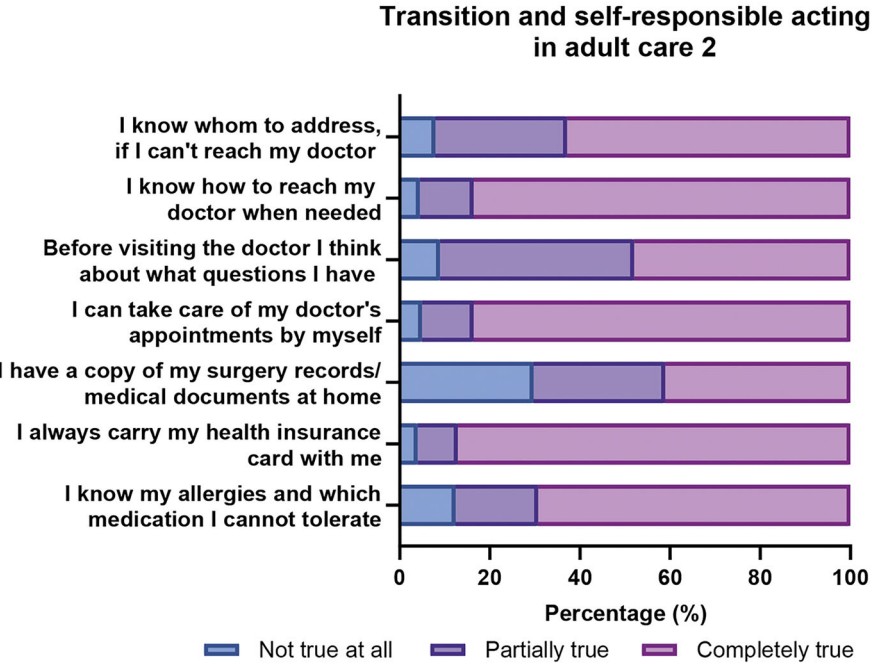

**Fig. 5 | Transition and self-responsibility of young adults with childhood onset obesity.** Depicted are the responses to a transition readiness assessment questionnaire as indicated by the graph labels. Responses were splitted into two panels (**A**) and (**B**) for better clarity (*N* = 202 subjects).

18–31 years and also lower shares of children with extreme obesity in relation to children with overweight or obesity than in our cohort[36].

Most of our participants had obesity complications such as (pre)diabetes, hypertension, elevated cardiovascular risk, and fatty liver disease. The overall prevalence of comorbidities was higher in our cohort (83.7%) as compared to the German general population: 33.8% of 18–29-year-olds and 40.9% of 30–44-year-olds reported health issues in the GEDA health survey[37]. In line, the prevalence of T2D was higher in our cohort than in the general German population up to 44 years (7.7% vs. <3.2%[37]). Similarly, we detected higher rates of arterial hypertension (24.5% vs. 4.9% in German

**Table 2 | Actual and desired transition of young adults with childhood onset obesity**

| A: How did the transition from childhood care to adulthood care take place? (*N* = 207) | *N* answers (percentage) |
|---|---|
| No transfer interventions | 136 (65.70%) |
| Concrete naming of contact persons after coming of age | 37 (17.87%) |
| Other (transfer to the family's general practitioner, support by family members) | 21 (10.14%) |
| Joint consultation with pediatrician and physician for adults | 7 (3.38%) |
| Flyers, brochures or technical information by pediatrician | 6 (2.9%) |
| Web-based transition platform | 0 (0%) |
| Structured transition program with several of the above mentioned elements | 0 (0%) |

| B: I would find the following aspects helpful (*N* = 54) | *N* answers (percentage) |
|---|---|
| Concrete naming of contacts after coming of age | 35 (64.81%) |
| Joint consultation with pediatrician and physician for adults | 30 (55.56%) |
| Training program (brochures, flyers, technical information) | 10 (18.52%) |
| Web based transition platform | 8 (14.81%) |
| Structured transition program with several of the above mentioned elements | 10 (18.52%) |

The table summarizes responses to an online questionnaire (*N* = 207 subjects), B refers only to those who desired a better transition (*N* = 54 subjects).

adults aged 18–29 years or 8.2% in adults aged 30–39 years)[38]. Regarding fatty liver disease, our rates were similar to the general European population (10.2% vs. 12.7%[39]). In contrast, a population-based German study found even higher shares of elevated liver enzymes, however looking at older subjects ranging 25–55 years[40]. Taken together, our results support the established link between childhood obesity and increased morbidity in midlife[13,41–45].

Regarding quality of life, more than half of our participants reported pain/discomfort (57.3%) and symptoms of depression and anxiety (52.0%). This is substantially higher than in the German general population, where 31.7% reported pain/discomfort and 17.9% reported anxiety/depression despite, a higher mean age of 47 years[46]. Also, the overall health rating in our cohort was lower when compared with the German population (74.4% vs. 93.5% for ages 20–29 years and 90.8% for ages 30–39 years[46]). Our findings are in line with results from a German study of youth and young adults with extreme obesity (age 14–24 years, BMI ≥ 30 kg/m²) who reported similarly low levels in quality of life that decreased with rising BMI from 74 to 70%[47]. Likewise, a representative study in Germany (age 16–97 years[48]) had higher levels of depression, anxiety and general health concerns among subjects with obesity compared to those with underweight, normal weight or overweight and Tian et al. revealed that higher BMI $Z$-scores from childhood to young adulthood were linked to lower physical health-related quality of life in mid-adulthood[49]. However, the high satisfaction with adult weight management even among weight gainers in our cohort suggests a complex relationship between physical and psychological well-being and health perception. This highlights the importance of considering both objective health measures and subjective perceptions of health beyond solely BMI measures in understanding and treating obesity. Based on those findings, we advocate for screening of physical and mental health conditions in young adults with obesity.

Nearly one-third of children with overweight/obesity did not continue regular healthcare visits and only a small minority stayed in specialized obesity outpatient care. This aligns with findings from other chronic diseases: 40% of young adults aged 19–25 years with type 1 diabetes dropped out of adult medical care compared to an 11% dropout rate for patients aged 18 years with support of a transfer navigator[21]. Similarly, 15% of patients with T2D received no care after transition, and 28% had not transitioned from pediatric care by a mean age of 21.4 years[22]. In this context, we recognized the crucial role of GPs as the primary contact for young adult patients, although addressing weight and obesity during visits happened seldom. This raises concerns whether patients and physicians may perceive discussions about weight-related issues as uncomfortable, disruptive, or inappropriate and whether GPs should receive training to effectively address obesity-related health concerns and play a more active role in treatment. In line, a global survey targeting obesity care in adolescence detected a misalignment between the perceptions of adolescence with obesity and health care professionals, e.g., regarding key motivators for weight loss, but still patients reported an overall positive feeling after weight loss discussion with a health care professional[50].

The Leipzig childhood obesity service initiates transition at age 17 years. Specialized obesity care for adults is located at the department of endocrinology at Leipzig University Medical Center, and we provide detailed contact information and brochures about this. In selected cases such as patients with bariatric surgery, severe complications or syndromic diseases joint consultations with the future treating physician are offered. Of note, not all children with obesity remain in the childhood obesity service at Leipzig University Medical Center until coming of age, as more than one third of adult participants reported they were last treated by a pediatrician at age 15 years or below. This might explain the overall low rates of specific transition interventions reported by the participants, although most young adults of our cohort experienced their transition as satisfactory. However, the latter should be questioned given the poor weight management outcomes and high rates of care discontinuation. Some expressed a desire for improvements, such as joint consultations with both pediatric and adult physicians and providing information for ongoing adult care, which would be simple to implement and time efficient. In line with that, many participants lost touch with specialized obesity care due to a lack of awareness about available options. Likewise, patients with type 1 diabetes reported negative experiences with transitioning in one third of the cases, especially criticizing the lack of preparation and poor information[51]. Sweeting et al.[52] studied the transition of adolescents with obesity qualitatively, noting that weight-related concerns were minimal in early adolescence but increased by the age of leaving school, suggesting late adolescence as a key intervention period. Overall, there is a need for interventions to enhance transition outcomes, underscoring the necessity of a structured transition process as per international guidelines[53,54]. We suggest to focus on primary medical care where most patients with obesity are seen and treated and where the most promising measures identified in our study would be easy to implement: e.g. setting up a structured transition plan with the patient, providing (written) information on disease related and healthcare system related issues, training primary healthcare providers in communicating and addressing weight-related concerns and complications. In this regard, Germany has recently implemented disease management programs for obesity in primary medical care[55], where those measures could be well

integrated. For children treated in secondary or tertiary care who might have a more complex history, additional measures like a transition navigator could be beneficial[21]. Therefore, joint efforts from healthcare professionals, healthcare providers, policy makers and patients are required.

As a strength, this study is the first to our knowledge that comprehensively addresses the transition process of obesity as a chronic condition. The longitudinal design allowed direct comparisons between childhood and adulthood. Moreover, data about diseases in early adulthood is generally scarce, with our study filling the gap during this crucial period. On the contrary, our study is limited by a high participant drop-out during follow-up, which may introduce selection bias. If we assume a positive selection bias given the generally low rates of health-seeking behavior in this population[47], the actual situation might be even more severe. However, we did not detect any systematic differences in BMI-SDS between those who finally participated in the follow-up study and those who were not willing to participate after successful recontact, although drop-out rates were higher among male participants (Table S1). Additionally, the single-center design restricts external validity and generalizability to the mainly urban Caucasian population around Leipzig. Moreover, some data were solely self-reported, thus limiting their reliability.

In conclusion, we confirmed the persistent health burden that children with overweight/obesity carry into adulthood, warranting regular screening for physical and psychological conditions in this patient group. We revealed discrepancies between actual and perceived health status related to obesity, which healthcare providers should recognize. Also, substantial gaps in the transition from pediatric to adult health care underscore the need for improving continuity and structured transition interventions in this vulnerable group.

## Data availability

The datasets generated and/or analyzed during the current study are available from the corresponding author, A.K., upon reasonable request. Summary source data for main Figs. 2–5 are provided in Supplementary Data 2.

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

## Acknowledgements

We would like to thank all patients and caregivers for participation in the study, as well as Caroline Schmieder for her help with patient recruitment. This work was supported by the German Research Foundation for the Clinical Research Center "Obesity Mechanisms" (grant CRC1052, project number 209933838, subproject C05) and project KO3512/3-1. Moreover, the project was funded by the Federal Ministry of Research, Technology and Space (Bundesministerium für Forschung, Technologie und Raumfahrt; BMFTR) as part of the German Center for Child and Adolescent Health (DZKJ) under the funding code 01GL2405A, and in addition with the grant 01GL1906 SUCCEED. The assessment of participants from the Leipzig Childhood Obesity Cohort, who reached adulthood during follow-up, was facilitated by the clinical trial unit of the Helmholtz Institute for Metabolic, Obesity and Vascular Research of the Helmholtz Zentrum München at the University of Leipzig and further supported with shared equipment from the LIFE child study. The LIFE child study is co-financed with tax funds based on the budget approved by the Saxon State Parliament (WI413). A.K., R.S., E.W. and J.R. acknowledge support by the German Diabetes Association (DDG). A.K. further acknowledges support by the JPI HDHL Metadis-funded CarbHealth consortium (01EA1908B) and the Novo Nordisk Foundation (NNF24SA0090441). R.S. and E.W. are supported by the German Research Foundation (Deutsche Forschungsgemeinschaft, DFG); project number 493646873—MD-LEICS. R.S. also acknowledges financial support by the Faculty of Medicine, University of Leipzig ("Nachwuchsförderprogramm").

## Author contributions

J.R. and R.S. designed the study, analyzed and interpreted the data and drafted the first version of the manuscript. J.R. and R.S. have directly accessed and verified the underlying data reported in the manuscript. N.G., K.Me., E.W., E.S. and R.S. participated in data acquisition, interpreted the analyses and revised the manuscript. K.Mü. and S.S. helped with study design and data interpretation and revised the manuscript. W.K., R.P. and M.B. acquired funding, helped with study design and data interpretation and revised the manuscript. A.K. designed the study, acquired funding, interpreted the results and revised the manuscript. All authors approved the final version of the manuscript, had full access to all the data in the study and accepted responsibility to submit for publication.

## Funding

## Competing interests

R.S. received honoraria from Novo Nordisk and Rhythm Pharmaceuticals. E.W. received honoraria from Novo Nordisk. M.B. received honoraria as a consultant and speaker from Amgen, AstraZeneca, Bayer, Boehringer Ingelheim, Daiichi-Sankyo, Lilly, Novo Nordisk, Novartis, and Sanofi. A.K.

received funding from the Novo Nordisk Foundation. All other authors declare no competing interests.

## Additional information

Johannes Riedel[1], Natascha Genge[1], Klara Meyer[1], Eric Wenzel [1,2,3], Elena Sergeyev[1], Katja Mühlberg[2,4], Sabine Steiner[2,5], Wieland Kiess[1,6], Roland Pfäffle[1,3], Matthias Blüher[2,7], Antje Körner [1,2,3,6,8] ✉ & Robert Stein [1,2,3,8]

[1]University of Leipzig, Medical Faculty, University Hospital for Children & Adolescents, Center for Pediatric Research Leipzig, Leipzig, Germany. [2]Helmholtz Institute for Metabolic, Obesity and Vascular Research (HI-MAG) of the Helmholtz Zentrum München at the University of Leipzig and University Hospital Leipzig, Leipzig, Germany. [3]German Center for Child and Adolescent Health (DZKJ), Partner Site Leipzig/Dresden, Leipzig, Germany. [4]University Hospital Leipzig, Department of Internal Medicine, Neurology and Dermatology, Division of Angiology, Leipzig, Germany. [5]Medical University Vienna, Department of Medicine II, Division of Angiology, Vienna, Austria. [6]LIFE–Leipzig Research Center for Civilization Diseases, University of Leipzig, Leipzig, Germany. [7]University of Leipzig, Medical Faculty, Department of Endocrinology, Nephrology and Rheumatic Diseases, Leipzig, Germany. [8]These authors contributed equally: Antje Körner, Robert Stein. ✉e-mail: Antje.Koerner@medizin.uni-leipzig.de

