## [Transparent Peer Review file · Communications Medicine]

A prospective cohort analysis from Germany shows transition into adulthood is an underestimated vulnerable period for children with overweight/obesity

Corresponding Author: Professor Antje Körner

Version 0:

Reviewer comments:

Reviewer #1

(Remarks to the Author)

Riedel et al. conducted a follow-up study on a cohort of children with overweight or obesity to examine their transition into adult healthcare, the challenges they faced during this process, and changes in their health status in adulthood. A total of 209 participants from the Leipzig region, Germany, agreed to take part in the study.

The authors report a high prevalence of obesity-associated comorbidities in young adulthood. Particularly striking was the observation that patients who had moved into a higher weight category reported that things went better with their weight during adulthood. Another noteworthy finding was how rarely participants recalled discussing their weight status with their physicians, raising further questions about patient-provider communication in this context.

Overall, the study addresses a highly relevant and important question. Publishing these findings is crucial to improving the transition of children and adolescents with obesity into adult healthcare.

Specific comments:

Abstract: In my opinion, the abstract does not yet adequately reflect the main results of the analysis.

Row 34: "Transition of children with overweight/obesity from childhood to adult care" would be preferable.

Row 36/37: "Mean age at follow-up in adulthood"

Row 38: Anthropometrics are missing.

Row 41: The first sentence is not a result of this analysis but refers to data from the literature. I do not understand the second sentence—what is it referring to?

Row 45: The range (18–87%) is very broad, and I do not understand how it came about. Either I was aware of a complication, or I was not.

Row 46: "The majority"—a numerical value is missing to contextualize this statement.

Row 48: "Better management"—Is this based on the results of self-developed questions in the questionnaire? This is not explicitly described in the results section.

Row 52: "Which is often not detected..."—This was not investigated here, as only patients were examined and surveyed, with no contact with treating physicians.

Introduction:

Row 69: "...by increasing rate of complications..." (please include the word "rate").

Methods:

Row 99: This section should be described more precisely, in my opinion. A total of 1,224 participants met the inclusion criteria, but only 209 ultimately participated.

Table 1:

A) Are there sociodemographic parameters from the last childhood examination, such as the highest level of education, that could be presented here?

B) It would be important to compare the characteristics of the adults who ultimately consented and participated with those of the adults who were successfully re-contacted at their last childhood examination. This could help investigate potential

selection bias.

Questionnaires:

I would appreciate if the developed questions were listed here. It would also be important to know who developed and analyzed these questions and by what method.

Findings:

Row 178: The authors should clarify that the 209 participants are those who consented to and participated in a follow-up examination.

Row 180: If I understand Figure 2 correctly, it represents the status of patients at the time of follow-up in adulthood. This is a cross-sectional analysis and does not allow conclusions about which patients remained in a specific weight category or changed categories between adolescence and adulthood. The described BMI dynamics seem more relevant here. The graphical results in Figure 2a–2c could also be shown in Table 1 or as supplementary material. Instead of Figure 3a, a more differentiated representation could illustrate which groups transitioned to which weight category or remained stable.

Rows 182–184: The described results do not match Figure 3a. The term "Maintainers" is used in the text, whereas "Stable," "Gainers," and "Losers" are used in the figure. I would prefer the term "Reduction in BMI category" over "Losers." Why did the authors use a different representation for Figure 3A compared to B–G? I would recommend using the format of Figures B–G.

Row 185: It would be important to clarify at this point which subsample (those with physical examination) the following statements refer to.

Rows 189–190: "Stable BMI" or "Stable BMI Category"? The authors use inconsistent wording, which makes reading and understanding the results section difficult. For example, "Weight loss" is mentioned here, whereas "Reduced BMI Category" was used in the previous section.

Row 200: The term "Improving BMI dynamics" appears here, introducing yet another wording variation. The terminology in Figure 3G should also be revised for consistency.

Rows 201–203: These examinations were only conducted in adulthood. What was the BMI distribution among patients with and without pathological IMT measurements? The result regarding carotid plaque formation is not shown. Please add the note "(not shown)."

Rows 210–211: It was not directly tested whether young adults assess chronic health burden differently than healthcare professionals, as only young adults were surveyed. I would suggest deleting this sentence.

Row 226: Was there a difference in the proportion of patients receiving ongoing care depending on whether they had overweight or obesity in childhood? It would also be interesting to examine differences between those whose category remained stable, increased, or decreased.

Row 246: Are these results based on the additional questions developed by the research team?

Row 267: This formulation is quite strong and, if applicable, only pertains to morbidity. The age distribution of patients (18–36 years) is listed in Table 1. Were there more patients over 30 years old than <30?

Rows 274–278: A comparison with another cohort that examined young adults with obesity, such as the YES cohort (Lennerz, 2018), would be more appropriate here.

Row 293: I believe it would be very important to reference the results of the ACTION Teens study in this part of the discussion.

Limitations:

The limitations of the study are mentioned in one sentence. However, they are not discussed in detail, nor are their implications for interpreting the results addressed. If we assume a positive selection bias, it is likely that the actual situation—especially for those who did not participate—is even more severe.

Table 1:

I suggest that the authors consider adding percentage values for comorbidities at different time points. This could potentially eliminate the need for Figure 2.

Figure 3:

Perhaps "Dynamics of glucose metabolism category: childhood vs. adulthood" would be a clearer title.

Figure 4:

Figure 4B: Is there information on when the patient last visited a doctor (specialized center, general practitioner)? There are likely differences between those receiving some form of care and those without any follow-up.

Figure 5:

Figure 5B: "How often is Overweight addressed?"—This likely refers to "Overweight or Obesity."

Figure 6:

There are many figures in this manuscript. Figures 6 and 7 are not well described in the results section. The authors should consider presenting these results in a table instead.

Figure 7:

Figure 7A: "Transition from childhood to adulthood"—The phrase "Transition from childhood care to adulthood care" would be clearer.

Reviewer #2

(Remarks to the Author)

This manuscript covers an important topic with very limited data which is transition of children with obesity to adult care, but they are issues that need to be clarified as below:

1. Can the authors add on what transitions plans the Leipzig childhood obesity service provides (used to provide) and comment in discussion in relation to answers to figure 5 C (“The majority did not know there was a service”) and Figure 7.
2. Do authors have data on time when the childhood obesity service plan transition as this will be valuable for the reader.
3. How long the children spent in the childhood obesity service as this might have affect transition?
4. At times the manuscript seems the manuscript deviates from the main aim that was described transition to report the presence of comorbidities – example of this is the very long paragraph of the discussion from lines 256 to 282 – please revise and use more data like from line 282 to 286.
5. Authors should expand on discussion about clinical implications of their results in relation to planning of transition services.

Other comments

1. Abstract should be revised and reflect what they did for example in background they did not characterized transition as there is no description of pathways there is a description of a cohort of adults who were seen early in a childhood obesity service. Similarly last part of the introduction (lines 84-86).
2. Plain language summary is not consistent with what authors report on manuscript or that need to be clarified. The sentence ... we accompanied children with obesity until they were young adults is not what is described in methods which is they contacted adults who used to be followed in the childhood obesity service.
3. General use consistent words for defining population at present have children, patient, population.
4. In results: Line 231 replace or seldom miss a dose for miss a dose maximum once per week as per figure.

Reviewer #3

(Remarks to the Author)

The study by Dr. Riedel and colleagues entitled “Transition into adulthood is an underestimated vulnerable period for children with overweight/obesity: a prospective cohort analysis from a German obesity center” aimed to describe the transition of children and adolescents with overweight or obesity into adulthood. It is already established that many children with obesity continue to have obesity in adulthood. However, how the transition from childhood obesity care to adult obesity care looks like has not been well-studied, and this prospective cohort analysis provides evidence to describe that transition. I found the study is well-written and has significance for clinical care and policy making. I have several comments and questions.

Questions regarding the study population:

1. Did the children included in the cohort have similar characteristics with children with overweight/ obesity in general? Or were they sicker (e.g., had a higher degree of obesity or higher risk for comorbidities)?
2. Did the study population receive regular treatment during childhood? If so, what kind of treatment and how long is the treatment? If they had received childhood obesity treatment and 93% of them had overweight/ obesity in adulthood, I wonder what this implied (was childhood obesity treatment ineffective?).
3. Is there any difference in characteristics between those willing and unwilling to participate in this study? While the authors have mentioned the possibility of selection bias, it would be great if this could be described further.

Other questions:

4. Does Germany have guideline for obesity care? It would be great to compare what should have been done vs the findings in this study to provide further understanding about the gaps. In Germany, is obesity care specialized, or do GPs also have competence and responsibility for treating it?
5. Do patients pay for obesity treatment? Does it contribute to the transition care?
6. To improve the transition of care, which actors are the most important one based on the findings? Is it the policy makers, clinicians, parents, other parties? It would be great if this is discussed more explicitly.
- 7) Were there any patients who received treatment with GLP-1 agonists or underwent weight loss bariatric surgery? I was thinking that those patients receiving treatment other than lifestyle modification might have longer contact with specialized care to adulthood.

Version 1:

Reviewer comments:

Reviewer #1

(Remarks to the Author)

Thank you very much for the thorough revision of the manuscript in response to the reviewers' questions, comments, and suggestions. The authors have addressed my remarks appropriately and provided well-reasoned responses. I continue to consider these findings important and valuable, and I support the publication of this work.

Reviewer #3

(Remarks to the Author)

Thank you for addressing the previous comments/ questions. I have no further comments.

Point-by-point response to the referees' comments

We appreciate the reviewers' thorough review and constructive feedback and will answer their comments as follows highlighted in blue.

Reviewer #1 (Remarks to the Author):

Riedel et al. conducted a follow-up study on a cohort of children with overweight or obesity to examine their transition into adult healthcare, the challenges they faced during this process, and changes in their health status in adulthood. A total of 209 participants from the Leipzig region, Germany, agreed to take part in the study. The authors report a high prevalence of obesity-associated comorbidities in young adulthood. Particularly striking was the observation that patients who had moved into a higher weight category reported that things went better with their weight during adulthood. Another noteworthy finding was how rarely participants recalled discussing their weight status with their physicians, raising further questions about patient-provider communication in this context.

Overall, the study addresses a highly relevant and important question. Publishing these findings is crucial to improving the transition of children and adolescents with obesity into adult healthcare.

We thank the reviewer for his/her positive feedback.

Specific comments:

Abstract: In my opinion, the abstract does not yet adequately reflect the main results of the analysis.

We have revised the abstract according to the reviewers' suggestions as outlined below. Particularly, we have put more emphasis on the discrepancy between the actual and perceived weight/BMI development during transition and how seldom weight status is discussed with the physician.

Row 34: "Transition of children with overweight/obesity from childhood to adult care" would be preferable.

We have corrected the wording as suggested.

Row 36/37: "Mean age at follow-up in adulthood"

We have corrected the wording as suggested.

Row 38: Anthropometrics are missing.

We have corrected the wording as suggested.

Row 41: The first sentence is not a result of this analysis but refers to data from the literature. I do not understand the second sentence—what is it referring to?

The first sentence refers to Figure 2A (now Table 1): All participants had overweight or obesity during childhood as defined by inclusion criteria of the study. Out of those, 93% still had overweight or obesity during adulthood.

The second sentence refers to Figure S2B. We have revised the sentence in the abstract accordingly:

“However, most patients (55.6%) who had moved into a higher BMI category during transition still perceived that things went better with their weight during adulthood.”

Row 45: The range (18–87%) is very broad, and I do not understand how it came about. Either I was aware of a complication, or I was not.

This sentence refers to the finding from Figure 4B (now Figure 3B): 30.3% of subjects with arterial hypertension, 86.7% of subjects with fatty liver disease and 18.8% of participants with (pre)diabetes were not aware of this condition before attending the outpatient clinical visit. We have changed the wording of the abstract for better clarity:

“87% of adults were not aware of one or more of these complications before.”

Row 46: "The majority"—a numerical value is missing to contextualize this statement.

We have added a numerical value as suggested.

“The majority (94.3%) lost contact to specialized obesity care...”

Row 48: "Better management"—Is this based on the results of self-developed questions in the questionnaire? This is not explicitly described in the results section.

This is based on self-developed questions. We have added more details as suggested to the results section: “Although 65.7% experienced lack of specific transition support (Table 2), the majority of participants (73.3%) did not express the need for improvement when being asked “Would you have preferred a better transition?”.

Row 52: "Which is often not detected..."—This was not investigated here, as only patients were examined and surveyed, with no contact with treating physicians.

We have changed to wording as suggested:

“Children with overweight/obesity carry a substantial health burden into adulthood which is often not perceived.”

Introduction:

Row 69: "...by increasing rate of complications..." (please include the word "rate").

We have corrected the wording as suggested.

Methods:

Row 99: This section should be described more precisely, in my opinion. A total of 1,224 participants met the inclusion criteria, but only 209 ultimately participated.

Table 1:

A) Are there sociodemographic parameters from the last childhood examination, such as the highest level of education, that could be presented here?

Unfortunately, there is no data on socioeconomic status during childhood available.

B) It would be important to compare the characteristics of the adults who ultimately consented and participated with those of the adults who were successfully re-contacted at their last childhood examination. This could help investigate potential selection bias.

We have added this analysis as suggested to the methods section:

“Only participants without syndromic or secondary obesity were included (N=1231) resulting in 209 subjects for further analysis (Figure 1, Table 1). Of note, patients who were not willing to participate after being successfully recontacted had a similar age- and BMI distribution at their last childhood visit than those who finally consented to the follow-up examination and slightly more men than women dropped out of the study (Table S1).”

Questionnaires:

I would appreciate if the developed questions were listed here. It would also be important to know who developed and analyzed these questions and by what method.

We have added this information to the methods section of the manuscript.

“Additional questions (Table S2) were developed by the research team through consensus of three pediatric experts.”

The analysis of questionnaires was done by summing up the different answering options checked by each participant as shares of the total number of participants. As we felt that this is clearly conveyed through the figure and table legends we did not specifically comment further on the analysis method.

Findings:

Row 178: The authors should clarify that the 209 participants are those who consented to and participated in a follow-up examination.

We have changed the wording as suggested:

“We conducted a follow-up of 209 patients (two thirds female) who attended the obesity outpatient clinic of Leipzig University Medical Center during childhood and consented to a follow-up examination at adulthood, accompanying their transition up to a maximum age of 36 years (Table 1, Figure 1).”

Row 180: If I understand Figure 2 correctly, it represents the status of patients at the time of follow-up in adulthood. This is a cross-sectional analysis and does not allow conclusions about which patients remained in a specific weight category or changed categories between adolescence and adulthood. The described BMI dynamics seem more relevant here. The graphical results in Figure 2a–2c could also be shown in Table 1 or as supplementary material. Instead of Figure 3a, a more differentiated representation could illustrate which groups transitioned to which weight category or remained stable.

As suggested, we have included the information of Figure 2 in Table 1 and used a different presentation of Figure 3A (now Figure 2A-B) with more emphasis on transition of BMI categories.

Rows 182–184: The described results do not match Figure 3a. The term "Maintainers" is used in the text, whereas "Stable," "Gainers," and "Losers" are used in the figure. I would prefer the term "Reduction in BMI category" over "Losers." Why did the authors use a different representation for Figure 3A compared to B–G? I would recommend using the format of Figures B–G.

As suggested, we have changed the presentation of Figure 3A according to the format of Figure 3B–G (now Figure 2C–H). We also changed the wording in the figure legends now consistently using „Reduction in BMI category“, „Maintenance of BMI category“ and „Gain in BMI category“ instead of „losers“, „stable“ and „gainers“.

Row 185: It would be important to clarify at this point which subsample (those with physical examination) the following statements refer to.

This information was already included in the manuscript:

“Regarding glucose metabolism after transition, more than one quarter of on-site study participants exhibited abnormal diabetes markers (Table 1) with 7.4% having overt diabetes.”

Rows 189–190: "Stable BMI" or "Stable BMI Category"? The authors use inconsistent wording, which makes reading and understanding the results section difficult. For example, "Weight loss" is mentioned here, whereas "Reduced BMI Category" was used in the previous section.

We changed the wording to consistent language as suggested.

“Of note, deterioration in glucose metabolism occurred more frequently among participants with maintenance or gain in BMI category than in those with a reduction in BMI category...”

Row 200: The term "Improving BMI dynamics" appears here, introducing yet another wording variation. The terminology in Figure 3G should also be revised for consistency.

We changed the wording to consistent language as suggested.

“We observed the strongest improvement of blood pressure in the group with a reduction in BMI category (43.9%)...”

Rows 201–203: These examinations were only conducted in adulthood. What was the BMI distribution among patients with and without pathological IMT measurements? The result regarding carotid plaque formation is not shown. Please add the note "(not shown)."

Patients with and without pathological IMT had a similar BMI distribution:

IMT classification	N participants	Mean BMI (kg/m²)	BMI range (kg/m²)
Pathological	113	35,9	19,7-65,7
Normal	16	36,6	22,9-55,7

Due to word count restrictions we did not add this information to the manuscript.

Furthermore, we have changed the wording of this section as suggested. Results of carotid plaque formation are included in table 1.

“Moreover, the vast majority of participants (82.2%) exhibited subclinical atherosclerosis during adulthood as represented by pathological IMT measurements and carotid plaque formation was detected in 5.4% of participants (Table 1).”

Rows 210–211: It was not directly tested whether young adults assess chronic health burden differently than healthcare professionals, as only young adults were surveyed. I would suggest deleting this sentence.

We changed the wording as suggested.

“Next, we tested whether the high and chronic health burden of young adults with childhood onset obesity is perceived adequately by the patients.”

Row 226: Was there a difference in the proportion of patients receiving ongoing care depending on whether they had overweight or obesity in childhood? It would also be interesting to examine differences between those whose category remained stable, increased, or decreased.

We conducted an additional analysis on dynamics of BMI category and continuation of care during adulthood and summarized the results in Table S2 and in the manuscript.

“Of note, patients with maintenance or a gain in their BMI category from childhood to adulthood were more likely to see a doctor on a regular basis than those with a reduction in BMI category (Table S2).”

Furthermore, there was no difference between those who received ongoing care and those who did not regarding the last recorded BMI-SDS during childhood (BMI SDS was 2.6 vs. 2.59).

Row 246: Are these results based on the additional questions developed by the research team?
Yes, these results are based on the additional questions developed by the research team.

Row 267: This formulation is quite strong and, if applicable, only pertains to morbidity. The age distribution of patients (18–36 years) is listed in Table 1. Were there more patients over 30 years old than <30? The majority of participants were below the age of 30 years at adulthood follow-up with an average age of 24.9 years.

Furthermore, we agree that our data does not allow conclusions about increased mortality risk and therefore changed the wording as suggested.

“Our results support the established link between childhood obesity and increased morbidity in midlife.”

Rows 274–278: A comparison with another cohort that examined young adults with obesity, such as the YES cohort (Lennerz, 2018), would be more appropriate here.

We have included the study to the discussion as suggested:

“Our findings are in line with results from a German study of youth and young adults with extreme obesity (age 14–24 years, BMI ≥ 30 kg/m²) who reported similarly low levels in quality of life that decreased with rising BMI from 74 to 70%.”

Row 293: I believe it would be very important to reference the results of the ACTION Teens study in this part of the discussion.

We have included the study to the discussion section as suggested:

“In line, a global survey targeting obesity care in adolescence detected a misalignment between the perceptions of adolescence with obesity and health care professionals e.g. regarding key motivators for weight loss, but still patients reported an overall positive feeling after weight loss discussion with a health care professional.”

Limitations:

The limitations of the study are mentioned in one sentence. However, they are not discussed in detail, nor are their implications for interpreting the results addressed. If we assume a positive selection bias, it is likely that the actual situation—especially for those who did not participate—is even more severe.

As suggested, we added more information to the discussion of limitations.

“If we assume a positive selection bias given the generally low rates of health seeking behavior in this population, the actual situation might be even more severe. However, we did not detect any systematic differences in BMI-SDS between those who finally participated in the follow-up study and those who were not willing to participate after successful recontact, although drop-out rates were higher among male participants (Table S1).”

Table 1: I suggest that the authors consider adding percentage values for comorbidities at different time points. This could potentially eliminate the need for Figure 2.

As suggested, we included the information of Figure 2 in Table 1.

Figure 3: Perhaps "Dynamics of glucose metabolism category: childhood vs. adulthood" would be a clearer title. We considered changing the title, however felt after doing so, that it was very long in relation to the figure and therefore decided not to change it.

Figure 4: Figure 4B: Is there information on when the patient last visited a doctor (specialized center, general practitioner)? There are likely differences between those receiving some form of care and those without any follow-up.

Unfortunately, we did not record information on when the patient last visited a doctor in further detail.

Figure 5: Figure 5B: "How often is Overweight addressed?"—This likely refers to "Overweight or Obesity."

We agree that the question targets overweight or obesity. However, in the survey we asked “On a regular doctor visit, is overweight addressed as a topic?” because most patients refer to overweight when they actually mean overweight and obesity.

Figure 6: There are many figures in this manuscript. Figures 6 and 7 are not well described in the results section. The authors should consider presenting these results in a table instead.

As suggested, we converted figure 7 into a table (Table 2).

Figure 7: Figure 7A: "Transition from childhood to adulthood"—The phrase "Transition from childhood care to adulthood care" would be clearer.

We changed the wording as suggested.

Reviewer #2 (Remarks to the Author):

This manuscript covers an important topic with very limited data which is transition of children with obesity to adult care, but they are issues that need to be clarified as below:

1. Can the authors add on what transitions plans the Leipzig childhood obesity service provides (used to provide) and comment in discussion in relation to answers to figure 5 C ("The majority did not know there was a service") and Figure 7.

We added this information to the discussion section.

"The Leipzig childhood obesity service initiates transition at age 17 years. Specialized obesity care for adults is located at the department of endocrinology at Leipzig University Medical Center and we provide detailed contact information and brochures about this. In selected cases such as patients with bariatric surgery, severe complications or syndromic diseases joint consultations with the future treating physician are offered. Of note, not all children with obesity remain in the childhood obesity service at Leipzig University Medical Center until coming of age as more than one third of adult participants reported they were last treated by a pediatrician at age 15 years or below. This might explain the overall low rates of specific transition interventions reported by the participants, although most young adults of our cohort experienced their transition as satisfactory."

2. Do authors have data on time when the childhood obesity service plan transition as this will be valuable for the reader.

See answer to comment 1.

3. How long the children spent in the childhood obesity service as this might have affect transition?

Unfortunately, we do not have sound data on this as times spent in the childhood obesity service vary greatly. Still, we asked participants about past weight loss attempts and added the information to the results section as below:

"Most young adults reported at least one weight loss attempt in the past (92.3%, Table S4) with dietary change on participants' own initiative being the most prevalent one (66.0%). Out of those, 37% achieved a reduction in BMI category (Table S4). Every fourth in our cohort (26.3%) had participated in a structured lifestyle intervention program in the past. Out of those, most participants maintained their BMI category (52.7%) and 32.7% reduced their BMI category (Table S4). Participants with bariatric surgery (N=8) either reduced (N=2) or maintained (N=6) their BMI category. All of them were seen by a doctor on a regular basis still during adulthood, 5 participants continued to receive specialized obesity care and 5 were seen by their GP (data not shown). Participants on treatment with a GLP-1 receptor agonist had mixed weight-loss responses with most of them maintaining their BMI category (N=3 out of 5). Again, all of them were seen by a doctor on a regular basis, either by a GP (N=3) and/or a diabetologist (N=3, data not shown)."

4. At times the manuscript seems the manuscript deviates from the main aim that was described transition to report the presence of comorbidities – example of this is the very long paragraph of the discussion from lines 256 to 282 – please revise and use more data like from line 282 to 286.

We have revised the discussion by shortening the section about comorbidities and expanding the section on quality of life as suggested.

"Taken together, our results support the established link between childhood obesity and increased morbidity in midlife. Regarding quality of life, more than half of our participants reported pain/discomfort (57.3%) and symptoms of depression and anxiety (52.0%). This is substantially higher than in the German general population where 31.7% reported pain/discomfort and 17.9% reported anxiety/depression despite a higher mean age of 47 years. Also, the overall health rating in our cohort was lower when compared with the German population (74.4% vs. 93.5% for ages 20-29 years and 90.8% for ages 30-39 years). Our findings are in line with results from a

German study of youth and young adults with extreme obesity (age 14–24 years, BMI \geq 30 kg/m²) who reported similarly low levels in quality of life that decreased with rising BMI from 74 to 70%.”

5. Authors should expand on discussion about clinical implications of their results in relation to planning of transition services.

We have expanded the discussion as suggested:

“Overall, there is a need for interventions to enhance transition outcomes, underscoring the necessity of a structured transition process as per international guidelines^{50,51}. We suggest to focus on primary medical care where most patients with obesity are seen and treated and where the most promising measures identified in our study would be easy to implement: e.g. setting up a structured transition plan with the patient, providing (written) information on disease related and healthcare system related issues, training primary healthcare providers in communicating and addressing weight-related concerns and complications. In this regard, Germany has recently implemented disease management programs for obesity in primary medical care, where those measures could be well integrated. For children treated in secondary or tertiary care who might have a more complex history, additional measures like a transition navigator could be beneficial. Therefore, joint efforts from healthcare professionals, healthcare providers, policy makers and patients are required.”

Other comments

1. Abstract should be revised and reflect what they did for example in background they did not characterized transition as there is no description of pathways there is a description of a cohort of adults who were seen early in a childhood obesity service. Similarly last part of the introduction (lines 84-86).

We have revised the abstract in parts. We also agree that the wording in regard to transition is somewhat confusing in general. However, we refer to “transition” in the background section of the manuscript as the “shift from pediatric care to adult medical care”. In our opinion this does not solely reflect specific transition measures and pathways (which we also describe) but also shifts e.g. in complications and healthcare access.

2. Plain language summary is not consistent with what authors report on manuscript or that need to be clarified. The sentence ... we accompanied children with obesity until they were young adults is not what is described in methods which is they contacted adults who used to be followed in the childhood obesity service.

We revised the plain language summary as suggested: “In our study, we conducted a follow-up of children with obesity at young adulthood.”

3. General use consistent words for defining population at present have children, patient, population.

We corrected the wording as suggested in several parts of the manuscript.

4. In results: Line 231 replace or seldom miss a dose for miss a dose maximum once per week as per figure.

We have corrected the wording as suggested.

“Still, participants with regular intake of medications (N = 92) reported on average a good adherence with 80.4% stating they miss a dose maximum once per week (Figure S5).”

Reviewer #3 (Remarks to the Author):

The study by Dr. Riedel and colleagues entitled “Transition into adulthood is an underestimated vulnerable period for children with overweight/obesity: a prospective cohort analysis from a German obesity center” aimed to describe the transition of children and adolescents with overweight or obesity into adulthood. It is already established that many children with obesity continue to have obesity in adulthood. However, how the transition from childhood obesity care to adult obesity care looks like has not been well-studied, and this prospective cohort analysis provides evidence to describe that transition. I found the study is well-written and has significance for clinical care and policy making. I have several comments and questions.

We thank the reviewer for his/her positive feedback.

Questions regarding the study population:

1. Did the children included in the cohort have similar characteristics with children with overweight/ obesity in general? Or were they sicker (e.g., had a higher degree of obesity or higher risk for comorbidities)?

In a recent representative study of the general German population between 3 and 17 years¹ 9.5% of children had overweight, 4.8% had obesity (without extreme obesity) and 1.1% had extreme obesity. Hence, when looking at

these three categories only, the shares of overweight, obesity and extreme obesity are 61.7%, 31.2% and 7.1%, respectively. The average BMI of children with obesity (defined as a BMI above the 97th percentile) in this study was 28.7 kg/m².

In comparison, our cohort comprised 12.0% children with overweight, 40.7% children with obesity and 46.4% children with extreme obesity. The average BMI of all children in our cohort was 31.4 kg/m².

Thus, we can assume that our cohort was more severely affected than children with overweight/obesity in general in Germany. We added this information to the discussion section of the manuscript:

“Of note, the KiGGS study, as a nation-wide German follow-up, reported slightly less obesity persistence rates between 37% to 50% among adults aged 18 to 31 years and also lower shares of children with extreme obesity in relation to children with overweight or obesity than in our cohort.”

2. Did the study population receive regular treatment during childhood? If so, what kind of treatment and how long is the treatment? If they had received childhood obesity treatment and 93% of them had overweight/ obesity in adulthood, I wonder what this implied (was childhood obesity treatment ineffective?).

We have added information on past weight loss attempts to the results section of the manuscript:

“Most young adults reported at least one weight loss attempt in the past (92.3%, Table S4) with dietary change on participants’ own initiative being the most prevalent one (66.0%). Out of those, 37% achieved a reduction in BMI category (Table S4). Every fourth in our cohort (26.3%) had participated in a structured lifestyle intervention program in the past. Out of those, most participants maintained their BMI category (52.7%) and 32.7% reduced their BMI category (Table S4). Participants with bariatric surgery (N=8) either reduced (N=2) or maintained (N=6) their BMI category. All of them were seen by a doctor on a regular basis still during adulthood, 5 participants continued to receive specialized obesity care and 5 were seen by their GP (data not shown). Participants on treatment with a GLP-1 receptor agonist had mixed weight-loss responses with most of them maintaining their BMI category (N=3 out of 5). Again, all of them were seen by a doctor on a regular basis, either by a GP (N=3) and/or a diabetologist (N=3, data not shown).“

Those findings highlight again the chronic nature of obesity, which we also added to the discussion section:

“In our study, the persistence of overweight or obesity from childhood to adulthood was high (93.3%) although 92% reported past weight loss attempts, supporting previous findings on the chronic nature of obesity^{5,6}.”

3. Is there any difference in characteristics between those willing and unwilling to participate in this study? While the authors have mentioned the possibility of selection bias, it would be great if this could be described further. We have added this analysis as suggested to the methods section:

“Of note, patients who were not willing to participate after being successfully recontacted had a similar age- and BMI distribution at their last childhood visit than those who finally consented to the follow-up examination and slightly more men than women dropped out of the study (Table S1).”

Other questions:

4. Does Germany have guideline for obesity care? It would be great to compare what should have been done vs the findings in this study to provide further understanding about the gaps. In Germany, is obesity care specialized, or do GPs also have competence and responsibility for treating it?

Germany has guidelines for obesity care in children (<https://register.awmf.org/de/leitlinien/detail/050-002>) and adults (<https://register.awmf.org/de/leitlinien/detail/050-001>). None of these guidelines mention the transition process, again emphasizing the need for more data on this topic. Furthermore, a guideline on “transition of young people with obesity from pediatric to adult medical care” has been registered in 2021 but has not been completed yet (<https://www.awmf.org/service/awmf-aktuell/transition-von-jungen-menschen-mit-adipositas-von-der-paediatric-in-die-erwachsenenmedizin>).

Regarding the second question, GPs in Germany have competence and responsibility for treating obesity. Their role has recently been strengthened through implementation of disease management programs for obesity in primary medical care. Specialized obesity care is offered in addition to that. We comment on this topic in the manuscript as indicated in our answer to question 6 (please see below).

5. Do patients pay for obesity treatment? Does it contribute to the transition care?

No, patients do not pay for obesity treatment in Germany (except for GLP-1 receptor agonists that are prescribed for the indication of obesity without type 2 diabetes).

6. To improve the transition of care, which actors are the most important one based on the findings? Is it the policy makers, clinicians, parents, other parties? It would be great if this is discussed more explicitly.

We have expanded the discussion as suggested:

“Overall, there is a need for interventions to enhance transition outcomes, underscoring the necessity of a structured transition process as per international guidelines^{50, 51}. We suggest to focus on primary medical care where most patients with obesity are seen and treated and where the most promising measures identified in our study would be easy to implement: e.g. setting up a structured transition plan with the patient, providing (written) information on disease related and healthcare system related issues, training primary healthcare providers in communicating and addressing weight-related concerns and complications. In this regard, Germany has recently implemented disease management programs for obesity in primary medical care, where those measures could be well integrated. For children treated in secondary or tertiary care who might have a more complex history, additional measures like a transition navigator could be beneficial. Therefore, joint efforts from healthcare professionals, healthcare providers, policy makers and patients are required.”

7) Were there any patients who received treatment with GLP-1 agonists or underwent weight loss bariatric surgery? I was thinking that those patients receiving treatment other than lifestyle modification might have longer contact with specialized care to adulthood.

We added this information to the results section of the manuscript:

“Participants with bariatric surgery (N=8) either reduced (N=2) or maintained (N=6) their BMI category. All of them were seen by a doctor on a regular basis still during adulthood, 5 participants continued to receive specialized obesity care and 5 were seen by their GP (data not shown). Participants on treatment with a GLP-1 receptor agonist had mixed weight-loss responses with most of them maintaining their BMI category (N=3 out of 5). Again, all of them were seen by a doctor on a regular basis, either by a GP (N=3) and/or a diabetologist (N=3, data not shown).“

References

¹ Schienkiewitz A, Damerow S, Schaffrath Rosario A, Kurth BM. Body mass index among children and adolescents: prevalences and distribution considering underweight and extreme obesity: Results of KiGGS Wave 2 and trends. *Bundesgesundheitsblatt - Gesundheitsforschung - Gesundheitsschutz* 2019;62:1225-34.